# WEAK AND STRONG GRADIENT DIRECTIONS: EXPLAINING MEMORIZATION, GENERALIZATION, AND HARDNESS OF EXAMPLES AT SCALE

## ABSTRACT

Coherent Gradients (CGH) is a recently proposed hypothesis to explain why over-parameterized neural networks trained with gradient descent generalize well even though they have sufficient capacity to memorize the training set. The key insight of CGH is that, since the overall gradient for a single step of SGD is the sum of the per-example gradients, it is strongest in directions that reduce the loss on multiple examples if such directions exist. In this paper, we validate CGH on ResNet, Inception, and VGG models on ImageNet. Since the techniques presented in the original paper do not scale beyond toy models and datasets, we propose new methods. By posing the problem of suppressing weak gradient directions as a problem of robust mean estimation, we develop a coordinate-based median of means approach. We present two versions of this algorithm, M3, which partitions a mini-batch into 3 groups and computes the median, and a more efficient version RM3, which reuses gradients from previous two time steps to compute the median. Since they suppress weak gradient directions without requiring per-example gradients, they can be used to train models at scale. Experimentally, we find that they indeed greatly reduce overfitting (and memorization) and thus provide the first convincing evidence that CGH holds at scale. We also propose a new test of CGH that does not depend on adding noise to training labels or on suppressing weak gradient directions. Using the intuition behind CGH, we posit that the examples learned early in the training process (i.e., "easy" examples) are precisely those that have more in common with other training examples. Therefore, as per CGH, the easy examples should generalize better amongst themselves than the hard examples amongst themselves. We validate this hypothesis with detailed experiments, and believe that it provides further orthogonal evidence for CGH.

## 1 INTRODUCTION

Generalization in over-parameterized neural networks trained using Stochastic Gradient Descent (SGD) is not well understood. Such networks typically have sufficient capacity to memorize their training set (Zhang et al., 2017) which naturally leads to the question: Among all the maps that are consistent with the training set, why does SGD learn one that generalizes well to the test set?

This question has spawned a lot of research in the past few years (Arora et al., 2018; Arpit et al., 2017; Bartlett et al., 2017; Belkin et al., 2019; Fort et al., 2020; Kawaguchi et al., 2017; Neyshabur et al., 2018; Sankararaman et al., 2019; Rahaman et al., 2019; Zhang et al., 2017). There have been many attempts to extend classical algorithm-independent techniques for reasoning about generalization (e.g., VC-dimension) to incorporate the "implicit bias" of SGD to get tighter bounds (by limiting the size of the hypothesis space to that reachable through SGD). Although this line of work is too large to review here, the recent paper of Nagarajan & Kolter (2019) provides a nice overview. However, they also point out some fundamental problems with this approach (particularly, poor asymptotics), and come to the conclusion that the underlying proof technique itself (uniform convergence) may be inadequate. They argue instead for looking at algorithmic stability (Bousquet & Elisseeff, 2002). While there has been work on analysing the algorithmic stability of SGD (Hardt et al., 2016; Kuzborskij & Lampert, 2018), it does not take into account the training data. Since SGD can memorize training data with random labels, and yet generalize on real data (i.e., its generalization behavior is data-dependent (Arpit

et al., 2017)), any such analysis must lead to vacuous bounds in practical settings (Zhang et al., 2017). Thus, in order for an algorithmic stability based argument to work, what is needed is an approach that takes into account both the algorithmic details of SGD as well as the training data.

Recently, a new approach, for understanding generalization along these lines has been proposed in Chatterjee (2020). Called the Coherent Gradients Hypothesis (CGH), the key observation is that descent directions that are common to multiple examples (i.e., similar) add up in the overall gradient (i.e., reinforce each other) whereas directions that are idiosyncratic to particular examples fail to add up. Thus, the biggest changes to the network parameters are those that benefit multiple examples. In other words, certain directions in the tangent space of the loss function are "strong" gradient directions supported by multiple examples whereas other directions are "weak" directions supported by only a few examples. Intuitively–and CGH is only a qualitative theory at this point–strong directions are (algorithmically) stable (in the sense of Bousquet & Elisseeff (2002), i.e., altered marginally by the removal of a single example) whereas weak directions are (algorithmically) unstable (could disappear entirely if the example supporting it is removed). Therefore, a change to the parameters along a strong direction should generalize better than one along a weak direction. Since the overall gradient is the mean of per-example gradients, if strong directions exist, the overall gradient has large components along it, and thus the parameter updates are biased towards algorithmic stability.

Since CGH is a causal explanation for generalization, Chatterjee (2020) tested the theory by performing two causal interventions. Although they found good agreement between the qualitative predictions of the theory and experiments, an important limitation of their work is that their experiments were on shallow (1-3 hidden layers) fully connected networks trained on MNIST using SGD with a fixed learning rate. In this work, we test CGH on large convolutional networks such as ResNet, Inception and VGG on ImageNet. While one of the tests of Chatterjee (2020) (reducing similarity) scales to this setting, the more compelling test (suppressing weak gradients by winsorization) does not. We propose a new class of scalable techniques for suppressing weak gradients, and also propose an entirely new test of CGH which is not based on causal intervention but on analyzing why some examples are learned earlier in training than others.

## 2 PRELIMINARY: REDUCING SIMILARITY ON IMAGENET

One test of CGH proposed in Chatterjee (2020) is to study how dataset similarity impacts training. Since directly studying similarity is difficult because which examples are considered similar may change during training (in CGH examples are similar if their gradients are similar), Chatterjee (2020) proposed adding label noise to a dataset based on the intuition is that no matter what the notion of similarity, adding label noise is likely to decrease it. Therefore, if CGH is true, we should expect that:

- As the label noise increases, the rate at which examples are learned decreases,
- Examples whose labels have not been corrupted (*pristine* examples) should be learned faster than the rest (*corrupt* examples), and,
- With increasing noise, since there are fewer pristine examples, the rate at which they are learned should decrease.

As preliminary experiment, we ran this test on ImageNet and the results for ResNet-18 are shown in Figure 1. The results for Inception-V3 and VGG-13 are very similar (please see Appendix A). We note the good agreement with the predictions from CGH thus providing initial evidence that CGH holds at scale.

## 3 ABLATING SGD TO TEST THE COHERENT GRADIENT HYPOTHESIS: SCALABLE TECHNIQUES TO SUPPRESS WEAK GRADIENT DIRECTIONS

Since weak directions are supported by few examples, CGH holds that overfitting and memorization in SGD is caused by descending down weak directions. The original CGH paper proposed to test CGH by modifying SGD to suppressing weak directions in order to verify that it significantly reduces overfitting (Chatterjee, 2020), i.e., improves generalization through greater algorithmic stability (Bousquet & Elisseeff, 2002).

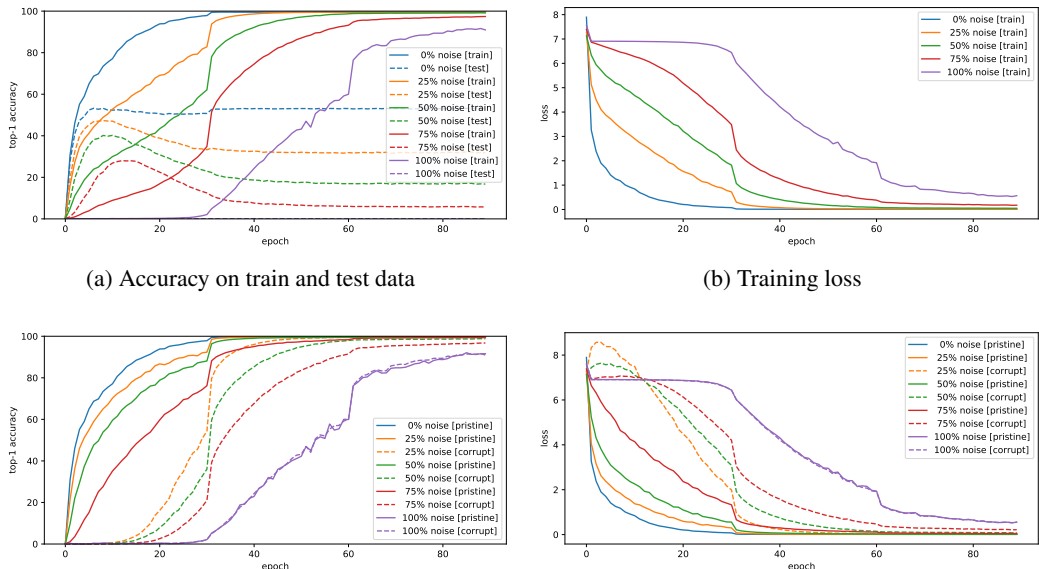

(a) Accuracy on train and test data     (b) Training loss

(c) Acc. on pristine and corrupt split of training data     (d) Loss on pristine and corrupt split of training data

Figure 1: Training curves of ResNet-18 without augmentation or weight decay on ImageNet dataset with various levels of noise in training labels. We randomize 0%, 25%, 50%, 75%, and 100% of the training labels to get 5 variants of ImageNet. The training examples whose labels are unaffected by randomization are called *pristine* and the rest are *corrupt*. We keep a fixed sample of 50k examples from the pristine and corrupt sets to measure our accuracy and loss (except for 100% noise where there are fewer than 50k pristine). Figures (a) shows the training and test accuracy with various levels of noise, while (b) shows the corresponding training loss. Figures (c) and (d) show the accuracy and loss plots on the pristine and corrupt samples. The plots confirm the predictions from CGH in Section 2. (We train with a mini-batch size of 256 and momentum of 0.9. The jumps at 30 and 60 epochs are due to the usual lowering of the learning rate by 1/10 from an initial value of 0.1.)

### 3.1 REVIEW OF THE WINSORIZATION TECHNIQUE

The test modified SGD by using the (coordinate-wise) *winsorized mean* of the per-example gradients in a mini-batch (instead of the usual mean) to update the weights in each step. Everything else, including learning rate was kept the same. The winsorized mean limits the influence of outliers by clipping them to a specified percentile of the data (called the *winsorization level*). As expected from CGH, they found that as the winsorization level increased, the rate of overfitting decreased.

We replicated this study with a ResNet-32 on CIFAR-10 and confirmed the results of the original study on this new dataset and architecture (please see Appendix B). We note that since the original study was only on MNIST which has low generalization gap, the effect of suppressing weak directions only manifested with label noise. On CIFAR-10 even the real label case (i.e., 0% noise) has significant overfitting which is reduced by winsorization. This provides stronger evidence in support of CGH.

However, a big challenge with winsorization is the need to compute *and store* per-example gradients which makes training much slower. For example, in our CIFAR-10 experiment we had to reduce the mini-batch size to 32 to make training feasible. This is exacerbated with larger models and more complex datasets such as ImageNet, and new techniques are needed to scale up the test.

### 3.2 TECHNIQUES BASED ON MEDIAN OF MEANS

We start by observing that the problem of suppressing weak gradient directions can be posed as a robust mean estimation problem from the robust statistics literature (Huber, 1981). Although winsorization is one way of obtaining robust mean estimates, there are others. In particular, the *median of means* algorithm (Minsker, 2013) is an optimal estimation technique in the sense that

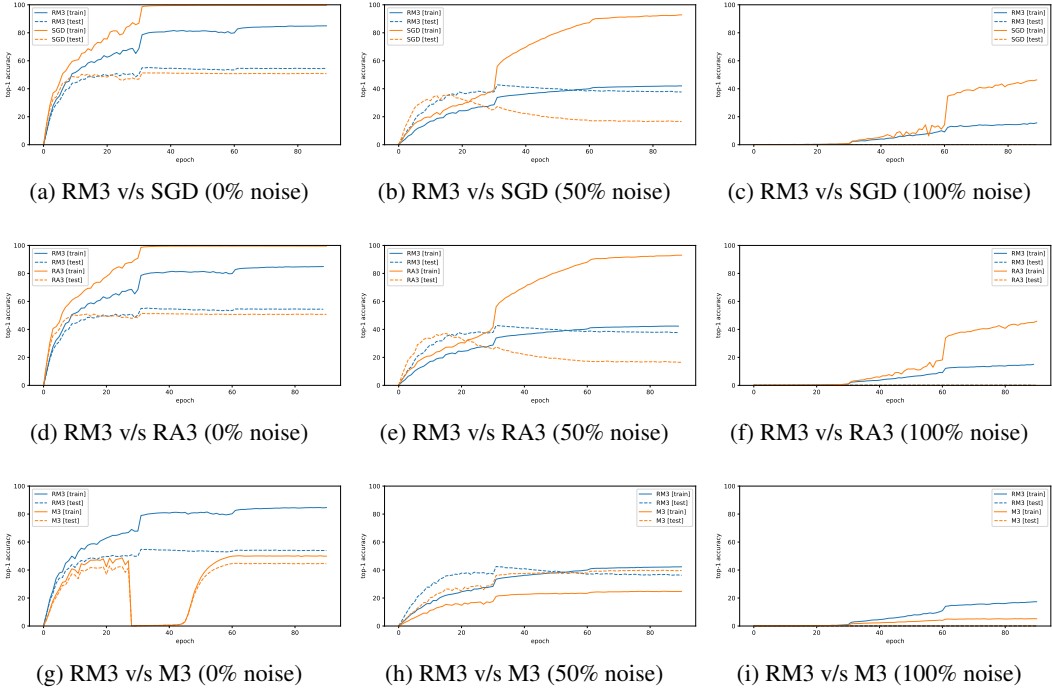

Figure 2: First row: Using the rolling median of 3 mini-batch gradients (RM3) to train a ResNet-18 on ImageNet leads to a smaller generalization gap than using SGD. Since this reduction in overfitting is obtained by suppressing weak gradient directions, this provides strong evidence for CGH in this setting. Second row: The use of median in RM3 is critical; using average instead (RA3) does not suppress (and resembles SGD). Last row: A comparison with M3, another method for suppressing weak gradient directions based on splitting a mini-batch into 3 groups and computing the median. We do not show error bars since the curves are on top of each other.

deviation from the true mean is bounded above by $O(1/\sqrt{m})$ with high probability ($m$ is the number of samples). The sample mean satisfies this property only if the observations are Gaussian.

The main idea of the *median of means* algorithm is to divide the samples into $k$ groups, computing the sample mean of each group, and then returning the *geometric median* of these $k$ means. The geometric median of $k$ vectors $x_1, x_2, \ldots x_k \in \mathbb{R}^d$ is the vector $y^*$ such that $y^* = \arg\min_{y \in \mathbb{R}^d} \sum_{i=1}^{k} \| y - x_i \|_2$. When $d = 1$, the geometric median is just the ordinary median of scalars. However, in high dimensions the algorithm to compute the geometric median (Weiszfeld, 1937) is iterative and is expensive to integrate in a traditional training loop. A simpler technique is to apply the *median of means* algorithm to each coordinate that gives a dimension dependent bound on the performance of the estimator.

**The M3 Technique.** The most obvious way to apply this idea to SGD is to divide a mini-batch into $k$ groups of equal size. We compute the mean gradients of each group as usual, and then take their coordinate-wise median. The median is then used to update the weights of the network.[1]

Even though the algorithm is straightforward, its most efficient implementation (i.e., where the $k$ groups are large and processed in parallel) on modern hardware accelerators requires low-level changes to the stack to allow for a median-based aggregation instead of the mean. Therefore, in this work, we simply compute the mean gradient of each group as a separate *micro-batch* and only update the network weights with the median every $k$ micro-batches, i.e., we process the groups serially.

---

[1]Since we are simply replacing the mini-batch gradient with a more robust alternative, this technique may be used to study optimizers other than vanilla SGD (such as SGD with momentum, ADAM, etc.). A systematic exploration of that is outside the scope of this study.

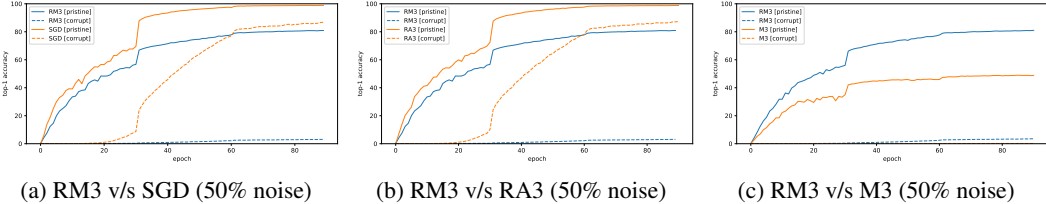

| (a) RM3 v/s SGD (50% noise) | (b) RM3 v/s RA3 (50% noise) | (c) RM3 v/s M3 (50% noise) |

Figure 3: The suppression of weak gradients by RM3 greatly inhibits learning of corrupt examples as compared to pristine. This shows that weak gradient directions are responsible for memorization and provides compelling evidence for CGH. We note again that median is what is critical, and not the rolling aspect since RA3 (Figure (b)) behaves like SGD (Figure (a)), but M3 (Figure (c)) like RM3. We do not show error bars since the curves are on top of each other.

In the serial implementation, $k = 3$ is a sweet spot. We have to remember only 2 previous micro-batches, and since $\text{median}(x_1, x_2, x_3) = \sum_i x_i - \min_i x_i - \max_i x_i$ (where $i \in \{1, 2, 3\}$), we can compute the median with simple operations. We call this *median-of-3 micro-batches* (M3).

**Example (Effectiveness of M3).** Fix $d$ and consider a set of $m < d$ training examples. At some point in training, let $g_i \in \mathbb{R}^d$ ($1 \leq i \leq m$) be their gradients. Suppose further that each gradient $g_i$ has an idiosyncratic component $u_i$ and a common component $c$, i.e., $g_i = u_i + c$ with $u_i \cdot u_j = 0$ (for $j \neq i$) and $u_i \cdot c = 0$. Since M3 involves coordinate-wise operations, let us assume that we are working in a basis where the non-zero coordinates of $u_i$ do not overlap with each other or with $c$.

Now, consider a mini-batch of size $3b \leq m$ constructed by picking $3b$ examples (i.e., their gradients) uniformly at random without replacement. The expected value of this update if we take an SGD step (i.e., simply take the mean of the mini-batch), is $g_{\text{SGD}} = c + \frac{1}{m} \sum_i u_i$. On the other hand, the expected value of the update with M3 is $g_{\text{M3}} = c$ since any non-zero coordinate of any $u_i$ cannot be the median value for that coordinate across the 3 groups since it can appear at most once.

In this extreme case, we see that M3 suppresses the weak gradient directions (the $u_i$) while preserving the strong gradient direction $c$.

**The RM3 Technique.** Now, rather than update the weights every $k$ micro-batches, as we do in M3, we can update the weights every micro-batch using the median from that micro-batch and the previous two. In this rolling setup, there is no longer a difference between mini-batches and micro-batches, i.e., this is essentially the same as computing the median over the current mini-batch gradient and the mini-batch gradients from the previous 2 steps. We call this *rolling median of 3 mini-batches* (RM3).

Two remarks concerning RM3 are in order. First, RM3 may be seen as an approximation to M3 chosen for implementation efficiency. The assumption is that the loss function does not change very much in 1 gradient step, i.e., it is locally stationary. Second, since RM3 uses recent gradient history, one may be tempted to think of RM3 as a form of momentum, but that would be wrong. We can understand this better by replacing the median operation in RM3 with the mean. We call this RA3 for *rolling average of 3 mini-batches*. As we shall see, it is the median v/s mean that makes a significant difference in suppressing weak gradient directions, not the rolling v/s non-rolling. Schematically, in their ability to suppress weak directions, we find that

$$\text{SGD} \approx \text{RA3} \ll \text{RM3} < \text{M3}. \tag{1}$$

### 3.3 Experimental Results

**Suppressing Weak Directions.** Our setup is similar to that of Figure 1. We use the ResNet-18 architecture and train on ImageNet with 0%, 50%, and 100% label noise with the same learning rate schedule, and no augmentation or weight decay. Unless otherwise specified, we use a mini-batch size of 256 and no momentum.

Figure 2 (a-c) shows the training and test accuracies of RM3 and SGD for the 3 noise levels. In all cases (including 0% noise, i.e, the real labels), we find that the suppression of weak gradients in RM3 leads to a lower generalization gap than SGD. The impact of suppressing weak gradients is even more clear when we look at the relative training accuracies of pristine and corrupt examples in the 50% label noise case. This is shown in Figure 3 (a). We see that unlike SGD which achieves high training

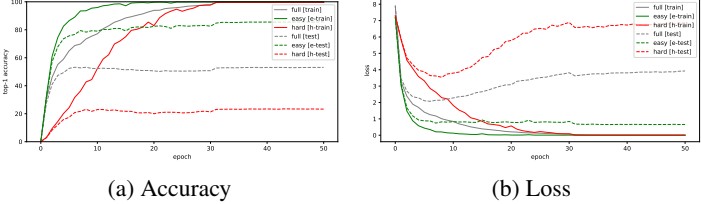

(a) Accuracy           (b) Loss

Figure 4: Using the original ImageNet training dataset, we create two datasets *easy* and *hard*. These are subsets of the original dataset and have a training set (*e-train* and *h-train*) and test set (*e-test* and *h-test*). We train on both with a ResNet-18 architecture. Figure (a) shows the accuracy curves of the full, easy and hard training sets evaluated on the original, e-test and h-test test sets. Figure (b) shows the corresponding training and test loss curves. The generalization of easy examples to other easy examples is better than that of hard examples to other hard examples, in accordance with CGH, and provides additional confirmation (Section 4.2).

| DIFFICULTY | RM3 ACCURACY | COUNT |
|:----------:|:------------:|:------:|
| 0 | 99.15% | 328400 |
| 1 | 97.82% | 100599 |
| 2 | 96.73% | 72814 |
| 3 | 95.69% | 63783 |
| 4 | 94.46% | 60389 |
| 5 | 92.85% | 62617 |
| 6 | 90.56% | 71495 |
| 7 | 86.61% | 99056 |
| 8 | 61.01% | 422014 |

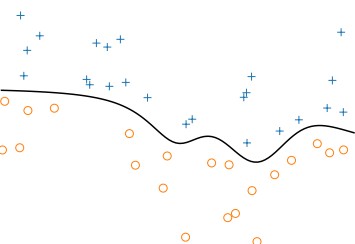

(a) Top-1 Training Accuracy of RM3 and Difficulty      (b) Decision Boundary and Hardness

Figure 5: (a) Training accuracy of RM3 on examples of varying difficulty. The suppression of weak gradients by RM3 disproportionately impacts the learning of more difficult examples which is as expected from CGH (Section 4.3). (b) If SGD enumerates hypotheses of increasing complexity then the examples learned early (i.e., easy examples) should be the ones far away from the decision boundary since they can be separated by simpler hypotheses. However, the hard examples then, should generalize well to the easy examples, but that is not what we find (Section 4.4).

accuracy on both pristine and corrupt examples, RM3 achieves good accuracy *only* on the pristine examples and does not learn the corrupt examples. Since suppressing weak gradient directions leads to a significant reduction in overfitting (and memorization), this *confirms the prediction from CGH that weak gradient directions are responsible for overfitting and memorization*, and increases our confidence in the theory.

**Ablation of RM3.** To ensure that this effect is due to suppressing weak gradient directions, and not due to the rolling nature of RM3, we compared RM3 with RA3 (Figures 2 (d-f) and 3 (b)) and with M3 (Figures 2 (g-i) and 3 (c))[2]. On the one hand, we see that there is a big difference between RM3 and RA3 and that the behavior of RA3 is almost identically to SGD. On the other, we find that M3 shows an even smaller generalization gap than RM3. (See schematic equation (1).)

**Remark: Algorithmic stability may come with optimization instability!** One unintended effect of using $\mathrm{median}$ to eliminate weak directions (and increase algorithmic stability) is that strong directions get amplified (c.f. $\mathrm{median}(0,0,1) < \mathrm{average}(0,0,1)$ with $\mathrm{median}(0,1,1) > \mathrm{average}(0,1,1)$). This may lead to larger steps in the strong directions which could increase loss (optimization instability). We see this for M3 with 0% label noise (Figure 2 (g)) where directions are expected to be stronger than in the 50% and 100% cases where this does not happen.

Note that mini-batch size for RM3 is important. The larger the mini-batch, the more it behaves like SGD (as may also be seen by considering the extreme case of full-batch training). Finally, RM3 increases the run-time of SGD by less than 1% whereas M3 adds about 30% overhead.

---

[2]We ran M3 with a batch size of 264 (i.e., micro-batch size of 88) instead of 256 since we use 8 GPUs and 264 is closest multiple of $3 \cdot 8$.

## 4 EASY AND HARD EXAMPLES

In this section, we study easy and hard examples through the lens of CGH. This is interesting for two reasons. First, we believe that this perspective leads to a better understanding of what determines the hardness of examples. Second, easy and hard examples provide a *fundamentally* different test of CGH than the label noise techniques used so far to test CGH since they do not force memorization.

### 4.1 BACKGROUND AND THE CGH PERSPECTIVE

Arpit et al. (2017) conducted a detailed study of memorization in shallow fully connected networks and small AlexNet-style convolutional networks on MNIST and CIFAR-10. One of their main findings is that for real data sets, starting from different random initializations, many examples are consistently classified correctly or incorrectly after one epoch of training which is in contrast to what happens with noisy data. They call these easy or hard examples respectively. They conjecture that this variability of difficulty in real data "is because the easier examples are explained by some simple patterns, which are reliably learned within the first epoch of training."

We believe that CGH provides an explanation of this phenomenon. But rather than say easy examples are explained by "simple" patterns (which begs the question of what makes a pattern simple), CGH would posit that easy examples are those that have a lot in common with other examples where commonality is measured by the alignment of the gradients of the examples. With this postulate it is easy to see why an easy example is learned sooner reliably: most gradient steps benefit it.

Note that this is a more nuanced phenomenon than claimed in Arpit et al. (2017). The dynamics of training (including initialization) can determine the difficulty of examples. In particular, it may explain the results on adversarial initialization (Liao et al., 2018; Liu et al., 2019) (where examples that are easy to learn with random initialization become significantly harder) and our own experience with anti-adversarial initialization (Section 2) since in both these cases the dataset remains the same (and thus the patterns remain the same).

### 4.2 TEST 1: EASY EXAMPLES SHOULD GENERALIZE BETTER THAN HARD EXAMPLES

If our hypothesis is true, the easy examples as a group have more in common with each other than the hard examples. Therefore, we would expect the gradients for the easy examples to be stronger than those for hard examples, and thus be more algorithmically stable (Bousquet & Elisseeff, 2002). In other words, they would be less impacted by the presence or absence of a single example. Therefore, we should expect that *the easy examples should generalize better than hard examples*.

To test this prediction, first we trained a ResNet-18 on ImageNet to 50% training accuracy using the normal training protocol. As per the discussion above, we call the examples that have been learned *easy* and the rest *hard*. Next, from the easy examples, we pick 500K examples and 100K examples at random (without replacement) to create a training set (*e-train*) and a test set (*e-test*). We train a ResNet-18 model from scratch (using the normal training protocol) on the e-train data and evaluate it on the e-test data to measure the generalization gap for the easy examples. Finally, we then repeat the process with the hard examples to get the generalization gap for the hard examples.

The results are shown in Figure 4. First, we verify that even in this setup of separate training, the easy examples (e-train) is learned faster than the hard examples (h-train). The slope of the corresponding losses early in training suggest that the gradients of the easy examples are more coherent than those of the hard examples (as per the argument in (Chatterjee, 2020, §2.2)). Finally, note that *the measured generalization gap for easy is significantly smaller than that for hard,* in accordance with the prediction above.

### 4.3 TEST 2: SUPPRESSING WEAK DIRECTIONS SHOULD IMPAIR LEARNING OF HARD EXAMPLES MORE

Based on the discussion above, we expect that the learning of hard examples is more reliant on weak gradient directions than the learning of easy examples. Therefore, if we use a technique to suppresses weak gradient directions during training with SGD, then we should expect that *the training accuracy on hard examples should be relatively worse than its training accuracy on easy examples*.

To test this, we extended the notion of easy and hard to a (slightly more) continuous notion of *difficulty*. To each example in the ImageNet training set, we associate a difficulty as follows. We run the easy/hard classification procedure (from Section 4.2) 8 times with different initializations. Since there is some variation from run to run in whether a particular example is classified as easy or hard, for each example, we count how many times it is classified as hard. We call this its *difficulty*. Thus, a difficulty of 8 means that in all 8 cases it was classified hard ("super hard"), whereas 0 means that in all 8 cases it classified easy ("super easy").

We then train a model for 90 epochs using RM3 on the ImageNet training set (as described in Section 3), and analyze how many examples at each difficulty level are correctly learned. Recall that *all* examples are correctly learned by SGD after 90 epochs. The results are shown in Figure 5(a). As expected, *the suppression of weak gradients makes it harder to learn the more difficult examples. For example, the super hard examples have a top-1 training accuracy of only 61% in comparison to 99% for the super easy ones.*

### 4.4 DOES SGD EXPLORE FUNCTIONS OF INCREASING COMPLEXITY? A POSSIBLE TEST.

One intuitive explanation of generalization is that SGD somehow explores candidate hypotheses of increasing "complexity" during training, thus finding the simplest hypothesis that explains the data. While there is some evidence backing this view (Arpit et al., 2017; Nakkiran et al., 2019; Rahaman et al., 2019), and this is not incompatible with CGH, we believe an aspect of our experiment suggests that this view is not entirely satisfactory.

In one interpretation of this view, as Figure 5(b) illustrates, one might think of the examples far away from the decision boundary as easy (since they can be separated by simpler hypotheses explored early on) and ones closer to the decision boundary as hard (since they need more complex hypotheses to be separated). One imagines that SGD in the course of training considers increasingly complex functions starting with perhaps linear classifiers (which fit many easy examples) and then refining it to increasingly complicated decision boundaries (to fit the remaining hard examples). From this, one might expect that *the decision boundary learned from the easy examples, generalizes poorly to the hard examples, but that learned from the hard examples generalizes well to the easy examples.*

We measure these quantities in the experiment of Section 4.2, and find them to be as shown below.

| model trained on | model evaluated on | accuracy | |
|---|---|---|---|
| easy (e-train) | easy (e-test) | 85% | shown as a baseline for good generalization |
| easy (e-train) | hard (h-test) | 17% | low (as expected) |
| hard (h-train) | easy (e-test) | 44% | low (unexpected) |
| hard (h-train) | hard (h-test) | 21% | shown for completeness |

Since the performance of the model learned from hard examples on easy test examples is much lower than that of easy-on-easy, one may be justified in saying that *hard examples do not generalize well to easy examples.* In other words, the examples learned late by SGD (i.e., the hard examples) by themselves are insufficient to define the decision boundary to the extent that the early examples are. This puts into question at least one interpretation of the viewpoint that SGD explores functions of increasing complexity during training.

## 5 CONCLUSION

Coherent Gradients provides an approach to understanding the data-dependent generalization observed in neural networks trained with SGD. In this paper, we have presented new evidence for CGH both through reproducing and scaling up previous work, and through new methods and experiments.

If CGH is true, our experiments with naturally occurring easy and hard examples suggest a subtle yet important shift in perspective: Generalization happens not because "simple" patterns are prioritized by SGD, but because common patterns are found first. (Of course, since common patterns are usually also simple, prevailing intuitions are not incorrect.)

For future work, it would be interesting to test CGH on other types of networks such as transformers as well as develop practical optimizers and regularizers based on techniques such as M3 and RM3.

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

## A    IMAGENET TRAINING DETAILS AND RESULTS ON INCEPTION AND VGG

We used PyTorch v1.3.1 and the implementations of the 3 models (ResNet-18, VGG-13, Inception-V3) used for ImageNet training are from TorchVision v0.4.2.[3] The initial learning rate for models was 0.1. For ResNet-18 we decreased it by a factor of 10 every 30 epochs (which is a default behavior of the example ImageNet trainer in PyTorch examples package) while for VGG-13 and Inception-V3 we used the cosine annealing schedule[4]. The Inception model did not use auxiliary logits. All experiments presented in Section 2 were run with momentum of 0.9 and a mini-batch size of 256. No random augmentation of the inputs or weight decay was used. To save compute we stopped VGG training early (when training accuracy reached 100%).

Figures 6 and 7 show the results of the label noise experiments on Inception-V3 and VGG-13 respectively. The results are essentially the same as that for ResNet-18 which has been presented in Section 2.

## B    WINSORIZATION ON CIFAR-10

We train a ResNet-32 on CIFAR-10 using SGD with a batch size of 32. We use a normal learning rate schedule where the initial rate is 0.1 and is lowered by $1/10$th first at 40K steps, and then every 20K steps thereafter. We train for a total of 100K steps (i.e., 64 epochs).

The amount of winsorization is controlled by a parameter $s$. First, we compute individual gradients for each example in a minibatch (this procedure is memory intensive for a large network like ResNet-32, which is why we use a smaller batchsize than usual). We perform winsorization by manipulating the per-example gradients in a coordinate-wise fashion. For the $i^{th}$ component, we collect the

---

[3]https://github.com/pytorch/vision/tree/master/torchvision/models
[4]https://pytorch.org/docs/stable/optim.html#torch.optim.lr_scheduler.CosineAnnealingLR

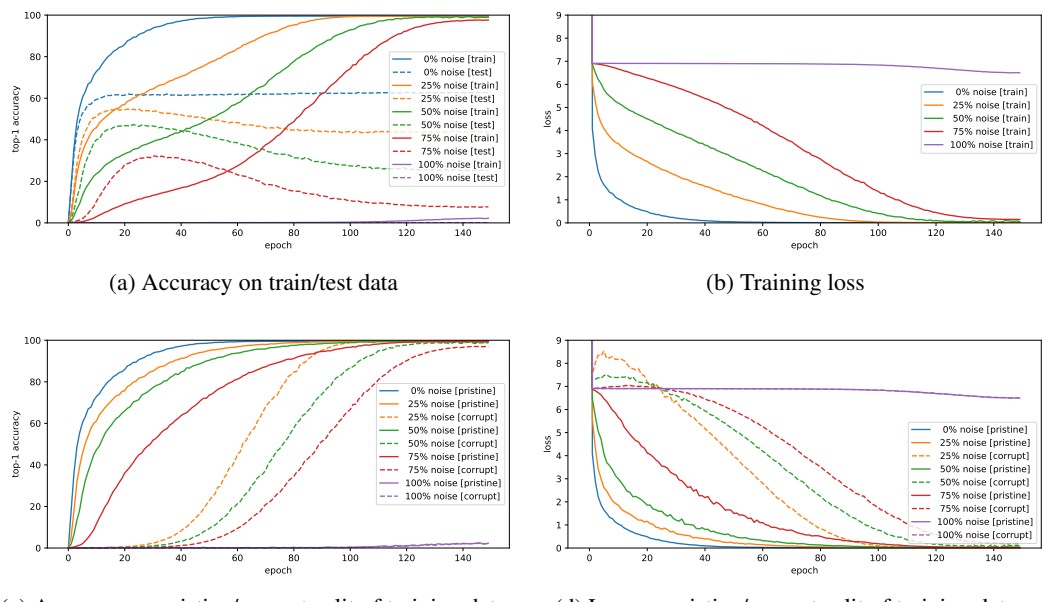

(a) Accuracy on train/test data

(b) Training loss

(c) Accuracy on pristine/corrupt split of training data

(d) Loss on pristine/corrupt split of training data

Figure 6: Training curves of Inception-V3 without augmentation or weight decay on ImageNet dataset with various levels of noise in training labels. We randomize 0%, 25%, 50%, 75%, and 100% of the training labels to get 5 variants of ImageNet. The training examples whose labels are unaffected by randomization are called *pristine* and the rest are *corrupt*. We keep a fixed sample of 50k examples from the pristine and corrupt sets to measure our accuracy and loss (except for 100% noise where there are fewer than 50k pristine). Figures (a) shows the training and test accuracy with various levels of noise, while (b) shows the corresponding training loss. Figures (c) and (d) show the accuracy and loss plots on the pristine and corrupt samples. The plots confirm the predictions from CG in Section 2.

per-example gradients to get 32 scalars. Instead of simply summing them to get the $i^{th}$ component of the overall gradient, we first reduce the impact of outliers by clipping each per-example scalar to be no less than the $(s + 1)$-smallest and no larger than the $(s + 1)$-largest of the 32 values, and then sum. Thus $s = 0$ corresponds to no clipping and thus regular SGD.

Figure 8 shows the performance of winsorization. The top row of the grid shows our results on the original dataset with winsorization parameter $s = 0, 2$, and $4$. We can see that as $s$ increases, the gap between train and test accuracy decreases. Also worth noting is that for small levels of winsorization, the model performance on test data does not change significantly. The bottom row in Figure 8 shows similar results when we add 100% label noise. At $s = 0$, we can see that the model completely memorizes the training data while generalizing very poorly. By increasing $s$ to 2 in order to suppress weak gradient directions, we find that the model completely fails to learn the training set. We view this as strong validation for CGH.

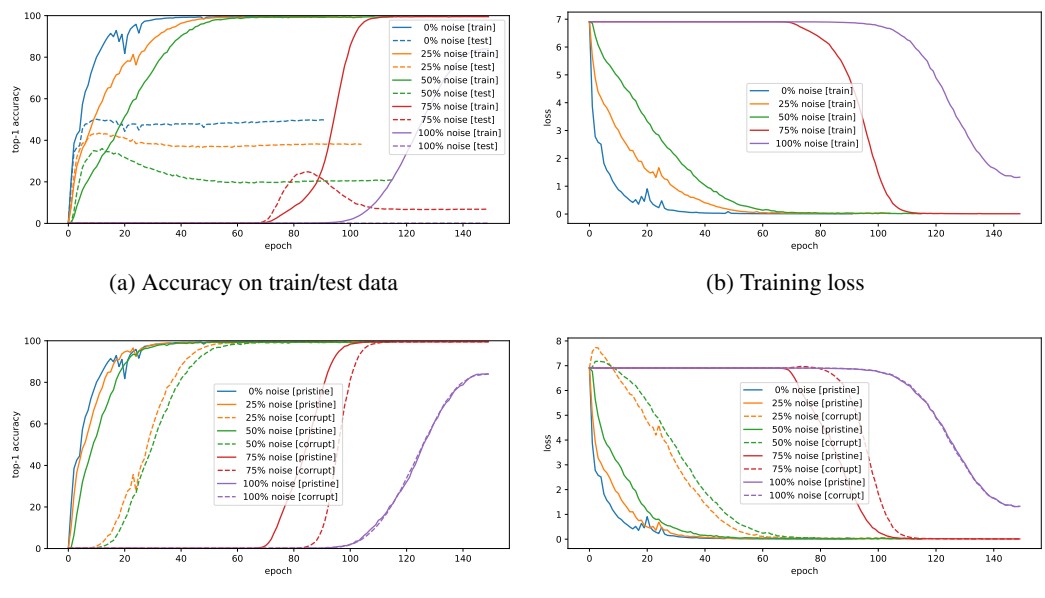

(a) Accuracy on train/test data

(b) Training loss

(c) Accuracy on pristine/corrupt split of training data

(d) Loss on pristine/corrupt split of training data

Figure 7: Training curves of VGG-13 without augmentation or weight decay on ImageNet dataset with various levels of noise in training labels. We randomize 0%, 25%, 50%, 75%, and 100% of the training labels to get 5 variants of ImageNet. The training examples whose labels are unaffected by randomization are called *pristine* and the rest are *corrupt*. We keep a fixed sample of 50k examples from the pristine and corrupt sets to measure our accuracy and loss (except for 100% noise where there are fewer than 50k pristine). Figures (a) shows the training and test accuracy with various levels of noise, while (b) shows the corresponding training loss. Figures (c) and (d) show the accuracy and loss plots on the pristine and corrupt samples. The plots confirm the predictions from CG in Section 2.

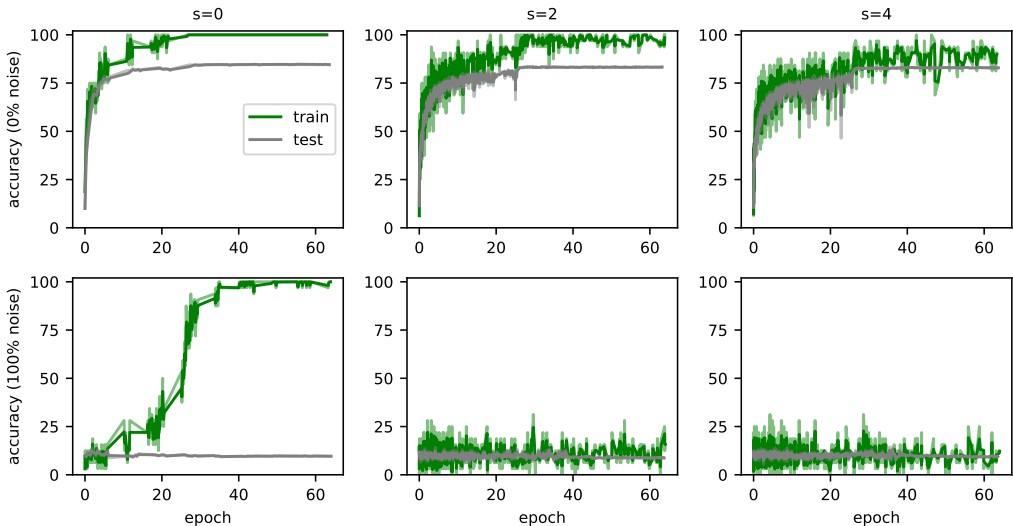

Figure 8: Performance of winsorization on the training of CIFAR-10 dataset on ResNet-32 architecture. The number of extreme samples in the mini-batch (of size 32) that are clipped are shown on the top of each column of the grid. The rows represent training with 0% and 100% label noise respectively. As the number of samples clipped by winsorization increases (i.e., more weak directions are suppressed), we can see that the generalization gap decreases. With 100% label noise and clipping 2 or 4 extreme samples, SGD does not memorize the training set.

