# OpenReview forum: "Weak and Strong Gradient Directions: Explaining Memorization, Generalization, and Hardness of Examples at Scale"
_ICLR.cc/2021/Conference — Reject_

### Official Review · AnonReviewer2 · 2020-10-28
**Nice experiments and foundation of a paper, but needs work in some baselines/conclusions and especially in organization/presentation.**

**Rating:** 5
**Confidence:** 5

**Review:**

**Summary of paper:**  Builds on and tests the "coherent gradients hypothesis" (CGH), which proposes that SGD is able to generalize well because of 'coherence' (similar direction) of gradients. The methodology of CGH involved comparing gradient directions on individual examples, infeasible for large datasets, so this work proposes and compares 2 mean- and median- based methods for testing CGH at scale. It also investigates easy/hard examples and thereby proposes a test of coherence of gradients based on the number of times an example of a certain difficulty is correctly classified.


**Pros/strong points:**
 - interesting experiments, solid relationships made to related work
 - locally clear explanations of techniques

**Cons/weak points:**
 - global flow/organization need a lot of work
 - builds heavily on previous work and is not sufficiently standalone
 - the main unique contributions of this work (separate from just validating CGH) are not sufficiently investigated to my understanding / satisfaction (see specific comments in "Quality" below)

**Summary of review + recommendation:** This has some nice experiments and I like to see papers building on and pushing further on previous insight (novelty is somewhat a weak point, but it is enough for me given the value of this kind of "building"). The presentation/organization needs a lot of work, and the main novelty proposed in this work (the methodologies that scale well) are missing an important baseline IMO. I also find some of the conclusions claimed (specifically sec.4.4) somewhat shaky. If these are addressed I am optimistic about increasing my score.


**DETAILED REVIEW:**

**Quality:**
 - The experiments are interesting and investigation of easy/hard examples is a nice addition
 - One of the main contributions of this work is methodology for computing similarity of gradients at scale (when it is infeasible to do so on a per-example basis as is possible e.g. for MNIST). A somewhat gaping hole in the testing of these methods then, to my mind, is to compare them to single-example computations: i.e.do experiments with RA3, RM3 and MA on minibatches of MNIST and CIFAR and make sure the conclusions are the same/agree with what is found via the single-example version.

**Clarity:**
 - A lot of sentences, especially in the intro, read very "conversationally" (lots of parentheses, asides, and relatively long sentences). This isn't bad, exactly, but it can be unclear especially for non-native english speakers, and is hard to parse quickly. I recommend reading over the whole text and trying to make the sentences shorter/punchier/each making only one clear point. Similarly for paragraphs, each should have a clear purpose/thesis in order to provide good flow and structure
 - Overall flow of the main part of the paper is very unclear; an "overview" or something at the end of the introduction could help a lot. Also confusing that methods are reviewed / terms defined in the experimental sections; maybe worth splitting these (but maybe having an overview would make it clear enough), and CGH isn't reviewed until after half of the experiments.

**Originality:**
 - The choices of experiments are logical for investigating CGH at scale, and the methods for doing so are novel to my knowledge
 - Relies heavily on previous ideas (CGH)

**Significance:**
 - Interesting experimental results, pushing forward our understanding/intuition of deep net generalization behaviour, but current presentation limits their significance

**Specific questions/recommendations:**

 - first sentence of abstract is a bit clunky, suggest rephrase as: The coherent grad hyp (CGH) was recently proposed to explain why....
 - Unclear flow in 2 sentences in abstract: In this paper ... Since ... | suggest rephrasing these sentences as "Since the techni ... new methods in order to validate CGH on ..."
 - suppressing weak gradient dirs: unclear link with generalization/memorization etc. or reason we would want to do this (can figure it out, but it's not clear in the abstract where it's important to be clear).
 - why is CGH causal? What were the interventions? They're referenced mid-paragraph without explanation.
 - seems like pristine examples should only be learned faster if the noise is sufficient to confuse the gradients (at low levels of noise, the noise can act as a regularizer and actually boost learning of mildly-noised samples)
 - winsorization experiments seem out of nowhere / don't flow or add insight (could cut most of this except for a line about the results in order to make space for the validation to single-example version suggested in the "Quality" section)
 - unclear what "increasing noise" means when introduced (needs 1-line explanation of how the label noise is generated).
 - Fig 5a) is not well explained/related to the text in the caption (doesn't mention what the count means, or that this is being proposed as a measure of coherence of gradients)
 - Fig 5b) how is this generated, is this just an illustration of what a decision boundary is or is this actual data? What data?
 - I get the intuition for the Remark on p6 but I find the explanation unclear (why do larger steps mean increased loss; I think an argument about variance might be more appropriate). I also had trouble parsing/don't understand the last sentence of the remark
 - background and CGH perspective should come before experiments on it
 - Why 50% acc for easy examples? Were other things tried? Arpit et. al used just one epoch.
 - What about the plots for generalization of h-train to e-test and e-train to h-test? I'd like to see these in appendix
 - Seems incorrect to conclude that "hard examples do not generalize well to easy examples" when the results show they generalize twice as well to easy examples (which are out of distribution) as to hard examples! These are still interesting results but the experiments would have to be more in depth (e.g. including visualizations, and small examples or something) to actually support this conclusion.
 - the paper overall reads like a sequel to CGH where you need to have read CGH to understand key plot points. Have someone who has not read CGH read it and point out places it does not stand well alone.

**Nits:**
 - "whereas directions that are idiosyncratic to particular examples fail to addup." wordy/unclear
 - " descending down weak directions." -> descending in OR toward (descending down is redundant and therefore confusing)
 - Figures need larger legends and maybe bolder lines- easy examples, generalizes poorly -> remove comma

---

### Official Review · AnonReviewer3 · 2020-10-28
**Work seems incremental**

**Rating:** 4
**Confidence:** 3

**Review:**

The paper gives algorithms to test the Coherent Gradients Hypothesis (CHG) for larger models and larger datasets. CGH is a recently proposed hypothesis to explain generalization of neural networks using algorithmic stability and other empirical observations for deep learning. It claims that there are strong gradient directions which are shared by many examples and those directions lead to better generalization where there is overfitting in weak gradient directions where only a few examples contribute. This paper extends the experiments to larger datasets like Imagenet and CIFAR and bigger models like ResNet, Inception, and VGG whereas the original paper looked at small fully connected networks for MNIST dataset. To test this hypothesis on larger datasets, they pose suppressing weak gradients as a robust mean estimation algorithm and propose two algorithms. This paper also gives another empirical test that can be used to confirm CGH.

1) The new algorithm which used the median of means technique to suppress weak gradients is interesting but I do not think that it is a significant enough contribution.

2) I find this paper’s perspective on easy and hard examples interesting that easy examples are those that have a lot in common with other examples and thus, are learned during the beginning of the training due to gradient alignment. They also give supporting experiments for this observation.

I would recommend rejecting this paper. I believe that the contribution of this paper is incremental in comparison to the paper which introduced CGH. This paper extends those experiments to larger datasets and models but I do not think that it is adding any significant new insight for the community.

Detailed Comments/Questions:

1) Using the earlier technique of coordinate wise winsorization, what is the largest batch size possible for Imagenet and does using a larger batch size significantly change the conclusions of this paper?
2) I am not entirely convinced by the arguments of section 4.4. This paper argues that learning only hard examples (which are learned later in the training period) does not generalize well to the easy examples and vice versa. Hence, if we think of hard examples as the ones close to the decision boundary, they should generalize well to the easy examples which is not true empirically. Hence, we can not fully think of SGD as learning functions of increasing complexity. I think more experiments are needed to fully understand this perhaps on smaller datasets to understand what is happening. Hard examples might not be sufficient to fully learn the training boundary but SGD still might be learning functions of increasing complexity.
3) I did not completely follow the comment on adversarial initialization in section 4.1 of the paper.

---

### Official Review · AnonReviewer4 · 2020-10-28
**Supporting Coherent Gradients Hypothesis on Scale**

**Rating:** 4
**Confidence:** 4

**Review:**

The paper aims at providing experimental evidence to support the Coherent Gradients Hypothesis (CGH), that was published previously. The hypothesis suggests that the ability of large neural networks to generalise comes from the aligned gradients of the examples in the dataset. Once SGD follows common gradients, ignoring the rare directions, the model will generalise better. For the experimental evidence the authors present two algorithms for approximate smoothing insignificant gradient directions for large scale networks and datasets, as opposed to the original CGH paper, where experiments were performed for MNIST and one hidden layer network. The proposed techniques allow for large-scale experiments and show a decreased generalisation gap compared to vanilla SGD. The authors also propose to analyse “easy” (ones that are learned first) and “hard” (require lots of training) examples. The claim is that the easy examples are the ones having coherent gradients, while hard ones push the network to rare directions.

The paper is clearly written, but contains some informal claims and formulations (more details in minor comments), that makes it difficult to follow the idea. The main goal of the paper is formulated as finding scalable evidences for CGH to be true, and the central part is the algorithms and experiments using them. The authors also proposed an “orthogonal” set of experiments, as compared to the original CGH paper, based on comparing generalisation on easy and hard examples. The authors provide the code and detailed descriptions of all the experiments.

The idea of CGH was already published, while the further step from my point of view should be trying to find more rigorous formal justification of similar gradients and their connection to the generalisation, rather than providing additional experiments, that are still just an indication that CGH might be true. If the proposed algorithms are considered to be a practical contribution for better generalising training, then the results should be compared to related techniques (for example, batch normalisation). Also, I would be highly interested in comparing the original winsorization approach to get rid of the insignificant gradient directions to the median robustness approach used here. Are they equivalent in the effect? Also, for easiness of computation, the algorithm calculates medians coordinate-wise - I wonder if from formal perspective this leads to something completely different than what was intended (since the original theory is introduced for the whole vectors up to my understanding). The discussion on the benefits of median compared to average reads too vague. Finally, the discussion on the complexity of learned network (section 4.4) is extremely informal and confuses more than it provides insights.

I would recommend the paper for rejection, because from my point of view it does not contain sufficient novel contributions for a conference publication.

Minor comments:

1 - sloppiness in SGD definition (as compared to mini batch GD): stochastic gradient descent is commonly used when the training is done without mini batches, i.e., updates are happening after each example seen

2 - sloppiness in gradient update definition (sum (in abstract) or average (later)?)

3 - algorithmic stability citation, Kuzborskij&Lampert - it is told they do not take training data into account, what does it mean? Their work is exactly about data dependent stability of SGD. I guess the problem is in wording - data dependency considers distributions, while dataset dependency considers properties of a particular dataset at hand.

4 - The claim that “any generalisation analysis that does not take into account dataset has to be vacuous” is somehow too strong. There are indications for it, but it cannot be claimed certainly.

5 - The gradients of examples are named to be a similarity measure between examples that does not change with training - but it does. Also, claim that label noise will always reduce any similarity measure between examples - what if only input part is taken into account for measuring similarity?

6 - the first experiment claims that hardship in learning label noise affected examples supports CGH - I cannot really agree with that. The added noise will make the network learn worse, that does not provide any evidence for aligned gradients being the reason for good generalisation. Moreover, in further experiments, one can see that vanilla SGD constantly outperforms the proposed methods on pristine examples (figure 3a) - does it mean that removing insignificatant directions hurts performance in some sense?

7 - figure1, c and d - are there any pristine examples when the level of noise is 100%?

8 - the first sentence in paragraph3.1 is poorly formulated

9 - it feels like the main point of the plots to all the experiments is to show the difference in final achieved accuracy/loss. If it is the case - why overload the reader with the full training development of the values? I would suggest to report it in more concise form.

10 - figure2 and 4 is extremely hard to read, too small

11 - easy and hard examples experiments: does the originally trained network, with vanilla SGD, generalise better on easy examples than on hard examples?

---

### Official Review · AnonReviewer1 · 2020-10-29
**Interesting work on CGH but analyses is not sufficient to posit CGH as the underlying explanation for generalization**

**Rating:** 4
**Confidence:** 5

**Review:**

### Pros

I appreciate an experimental study to delve deeper into the dynamics of CGH. The RM3 algorithm is very interesting and scalable and can perhaps be adapted to be used more regularly for optimization in the presence of noise.

### Cons

In short my main issue is as follows - The entire idea of the paper is to present empirical evidence to support CGH as the explaining factor behind generalization. However, if there are a lot of confounding variables in the experimental setup, in my opinion, it becomes very hard to draw reasonable conclusions from the experiments. There are also multiple other papers (which is too large to list here) that propose different explanations for generalization and mitigation techniques to improve generalization. So, I believe that if one were to posit one of the theories over the other theories it is important to show specific situations where the other explanations fail and this succeeds. I do not see any such experiments in this paper.

In addition, it is not a theoretical paper (which is absolutely alright in my opinion) but being an empirical paper its experiments need to be thorough and convincing enough so that some conclusion can be drawn out of it and I think it would be wrong to draw reasonable conclusions out of the experiments in this paper in its current form.

Perhaps, most importantly, the paper seems to suggest that the only way to generalization, and which is supported  by CGH to a certain extent, is to not memorize label noise in noisy datasets. This seems to be at odds with the behaviour where neural networks can generalize even in the presence of label noise while memorizing it (*benign overfitting*). The authors do not discuss theoretical and practical works that discuss this phenomenon of benign overfitting or the importance of memorization for learning in neural networks.

#### Detailed points

If the key idea is that it is CGH that is causing generalization and not something else, the paper must compare itself with other regularizers and training techniques that claim to improve generalization and either explain why CGH is not at odds with them or show experimental or theoretical scenarios where CGH succeeds and others fail. For something, which is as purely experimental and intuitive as CGH, I think the experiments need to be more through than what it is. The benefits of generalization could have easily been brought about by something else other than the robust mean estimation in RM3, which would completely negate the hypothesis.

The three points proposed in  Section 2 (which is a preliminary argument as admitted by the authors) are not something typical of CGH. While CGH could be a possible explanation, they are too generic to be very specific about CGH. It could easily be about learning progressively complex hypotheses.


In Fig 1c, the accuracy of  _pristine_ examples increases faster for datasets with more label noise, which agrees with  Point 1 in Section 2. However, I am not sure how the fact that _corrupt_ examples also see a similar trend i.e. noisy examples in datasets with more noise take longer to learn support this argument. As all noisy examples are uniformly randomly labelled, shouldn't they all see equally _randomly_ _oriented_ gradients and thus learn at the same rate ?

Some experiments show _underfitting_, which might explain why it is showing better generalization in the presence of label noise. For example, experiments with 50% label noise in Fig 2, show a train accuracy of around 40% which is quite low. One possible explanation is that the hyper-parameters used here are preventing further optimization and thus memorization of the noisy labels (and the authors do state that they use the original hyper-parameters and do not tinker with them).  Thus, it cannot be said with certainity that the cause of generalization in those experiments is due to RM3 and not due to the particular choice of hyper-parameters.

Further, as I mentioned before, neural networks can exhibit generelization even when memorization is exhibited.

I also find the experiments in Sec 4.4 to be very unsatisfactory.  One of the hypothesis that this section is based on is that - hard examples are the ones closer to the boundary. That would mean that harder examples are ones that are confusing and it can belong to either of the two classes. However, that is not necesasarily true. Hard examples can just be rare examples  not necessarily close to the boundary. This would also support both the stats in the table in 4.4. corresponding to a) h-train, e-test b) h-train, h-test without being at odds with "SGD learns functions of increasing complexity".


The authors have not prepared a rebuttal.

---

### Decision · Program_Chairs · 2021-01-07
**Final Decision**

**Decision:**

Reject

**Comment:**

The reviews are concerned about the novelty/incremental nature of the paper and partially also about the
conclusions drawn from the experiments. The authors did not take the chance to write a response.